# Response of Grassland Soil Quality to Shallow Plowing and Nutrient Addition

**DOI:** 10.3390/ijerph20032308

**Published:** 2023-01-28

**Authors:** Bin Li, Guohua Ren, Xiangyang Hou, Xiaotian An, Guanhua Lv

**Affiliations:** 1College of Grassland Science, Shanxi Agricultural University, Jinzhong 030801, China; 2Youyu Loess Plateau Grassland Ecosystem National Research Station, Shanxi Agricultural University, Jinzhong 030801, China; 3Shanxi Key Laboratory of Grassland Ecological Protection and Native Grass Germplasm Innovation, Jinzhong 030801, China; 4Key Laboratory for Model Innovation in Forage Production Efficiency, Ministry of Agriculture and Rural Affairs, Jinzhong 030801, China

**Keywords:** grassland, shallow plowing, nutrient, soil microbial quotient, soil quality, principal component analysis

## Abstract

Due to differences in the soil environment and grassland management measures, the change characteristics of soil microbial entropy and soil comprehensive quality in different types of grassland may vary significantly. In this study, the spatial variation characteristics of grassland soil microbial entropy under shallow plowing and nutrient addition measures were studied using a networking experimental platform established in a temperate meadow steppe, typical steppe, and desert steppe in northern China. The grassland soil quality was comprehensively evaluated to provide a theoretical basis for the scientific and reasonable management of grasslands under global climate change. The results show that in the meadow steppe, shallow plowing and nutrient addition significantly decreased the soil microbial biomass carbon and microbial entropy in the region, resulting in a decrease in the comprehensive score of soil quality. In the typical steppe, due to the influence of shallow tillage measures, the microbial biomass of the grassland soil in the region was higher than that of the control group and its two treatments, and the comprehensive score of soil quality was ranked first among the four treatments. In the desert steppe, the interaction of shallow plowing and nutrient addition significantly increased the soil microbial entropy in the region. Under the nutrient addition measures, the soil microbial entropy of the desert steppe showed a downward trend. In addition, the soil C/N ratio of the desert grassland under nutrient addition measures increased significantly, and the comprehensive score of soil quality ranked first among the four treatments as the microbial entropy decreased significantly.

## 1. Introduction

With the concepts of “carbon neutralization” and “carbon peak” being proposed, the change characteristics of the carbon cycle under global climate change have become a topic of interest in academia [1]. Grasslands are important ecological resources in China, accounting for 41.7% of China’s land area and exceeding the total areas of farmland and forest. Grasslands are not only the largest terrestrial ecosystem in China but also one of the most important terrestrial carbon sinks [2]. However, in recent years, the degradation of grassland ecosystems has become increasingly serious under the combined effects of natural and human factors, which include global climate change and overgrazing [3]. Simplification of the grassland community structure and frequent weakening of ecosystem function have significantly affected grassland carbon input [4]. In terms of the improvement and restoration of degraded grassland, shallow plowing and nutrient addition are widely used management measures. Zhang found that shallow plowing could optimize soil water, fertilizer, air, and heat conditions in the improvement of degraded *Leymus chinensis* grassland [5]. Studies have shown that shallow plowing measures can promote the growth and metabolism of beneficial microorganisms in the soil and the decomposition of organic matter, thus promoting plant growth and improving the productivity of grassland communities [6,7]. Nutrient addition measures can significantly improve soil fertility [8]. In a study of the Songnen Plain in Heilongjiang Province, Liu found that the pasture output could be efficiently improved by treatments of fertilization and irrigation [9]. Ma suggested that plowing would affect the activation of soil seed banks [10]. There is no denying that shallow plowing and nutrient addition measures are important in ameliorating soil quality and improving the sustainable development of grassland ecosystems.

Soil microorganisms are not only decomposers and consumers in grassland ecosystems but also important participants in the soil carbon cycle; as such, they are one of the most important biological driving forces in soil [11]. As the most active part of soil, the numbers and activities of soil microorganisms will also change due to plowing, fertilization, and other management measures. Studies have shown that deep plowing can significantly increase soil microbial quantity [12], and ridge plowing can increase the numbers of bacteria, fungi, and actinomycetes in the soil [13]. Jing Jingying et al. showed that exogenous nutrient addition could significantly reduce the relative abundance of phagotrophic protozoa in the soil and further affect the structure and function of the soil ecosystem mediated by microorganisms [14]. Fertilization can significantly increase the relative abundance of beneficial microbial flora in soil and promote the accumulation of soil microbial biomass [15]. In a study of semi-arid grassland soils, nitrogen fertilization increased the β-diversity of soil bacteria (rather than fungi) [16] and the carbon sequestration capacity of soil microorganisms [17]. Soil microbial biomass carbon (MBC) is the active part of soil organic matter that represents the carbon contents of bacteria and fungi cells per unit of dry soil and is an effective indicator of soil biological activity and nutritional status. In addition, soil microbial carbon is a key component of the carbon pool that can feed back on climate change [18]. Soil microbial entropy (qMB) is the ratio of soil MBC to soil organic carbon (SOC) that can be used to indicate changes in soil quality [19]. Therefore, soil microbial entropy can be used as an effective evaluation index for grassland soil quality under different grassland management measures.

The temperate grassland region spans a large area in northern China, and there are various types of grassland resources distributed from west to east [20]. Grassland forms a vital ecological barrier in China, and its unique ecological location has resulted in its sensitivity to climate change [3]. In recent years, under the influence of global climate change and human disturbance, the region has faced serious degradation. Shallow plowing and nutrient addition are common grassland management measures in the region. At present, studies on the effects of shallow plowing and nutrient addition on grassland ecosystems have generally focused on plant growth and physiological characteristics [21], productivity and community composition [22], soil physical properties [7,23], the soil seed bank [24], and microbial diversity [25,26]. Soil nutrient status and soil quality can be effectively understood by measuring the soil microbial entropy in grasslands [19], and this measure has significance for the systematic and rational management of grassland under global climate change. However, the spatial variation characteristics of grassland soil microbial entropy under shallow plowing and nutrient addition have not been reported.

Here, we studied the effects of shallow plowing, nutrient addition, and their interaction on the grassland soil quality of three typical grasslands in order to reveal the influencing mechanisms of shallow plowing and nutrient addition on soil quality at the regional scale using an experimental platform established in northern China in 2019 that comprises a temperate meadow steppe, typical steppe, and desert steppe. Our hypotheses were as follows. Hypothesis 1: Shallow plowing and nutrient addition measures will change the original soil structure of the grassland and will have an impact on the soil microbial community. Microbial entropy will also show different responses to shallow plowing and nutrient addition measures in different grassland types. Hypothesis 2: In different types of grassland, soil quality will be significantly changed by shallow plowing and nutrient addition measures.

## 2. Materials and Methods

### 2.1. Study Area

This experiment was conducted in the shallow plowing and nutrient addition networking experimental platform established in temperate grasslands in northern China (Figure 1). The ecosystem types included a meadow steppe, typical steppe, and desert steppe located at the Hulunber Grassland Ecosystem Observation and Research Station, Institute of Agricultural Resources and Regional Planning, Chinese Academy of Agricultural Sciences; the Youyu Loess Plateau Grassland Ecosystem National Research Station, Shanxi Agricultural University; and the Urat Desert–Grassland Research Station, Northwest Institute of Eco-Environment and Resources, Chinese Academy of Sciences, respectively. In 2019, an experimental platform was established by selecting relatively homogeneous vegetation and soil at each site and typical representative sections, and all sampling was performed in accordance with common experimental protocol.

The Hulunber Grassland Ecosystem Observation and Research Station, Institute of Agricultural Resources and Regional Planning, Chinese Academy of Agricultural Sciences (49°19′349″ N, 119°55′52″ E), at an altitude of 628 m, has a temperate continental monsoon climate. The annual average temperature is 1 °C, the maximum temperature is 20.49 °C, the minimum temperature is −36.4 °C, the average annual precipitation is 354.92 mm and concentrated in July−September, and the frost-free period is about 110 d. The soil type is chernozem or chestnut soil, gradually transitioning from east to west, from a chernozem zone to a chestnut soil zone. The hidden soil includes meadow soil, swamp soil, and saline–alkali soil. The soil has an excellent aggregate structure and a high organic matter content. The zonal vegetation type is *Leymus chinensis* + forb meadow steppe. The Youyu Loess Plateau Grassland Ecosystem National Research Station, Shanxi Agricultural University (38°59′48.5″ N, 112°19′39.6″ E) is at an altitude of 1384 m. The area has a temperate semi-arid climate, with an average annual temperature of 4.7 °C, a maximum temperature of 25.8 °C, a minimum temperature of −17.8 °C, and an average annual precipitation of 435 mm. Precipitation is concentrated in July−September, and the frost-free period lasts for 100–120 d. The soil type is dry, loose, soda–alkaline soil, with vertical joint development. The soil is highly saline, and spring drought occurs frequently. The zonal vegetation type is typical steppe of *Leymus secalinus* + forbs. The Urat Desert–Grassland Research Station, Northwest Institute of Eco-Environment and Resources, Chinese Academy of Sciences (41°25′15″ N, 106°58′11″ E) is located at an altitude of 1670 m and has a typical temperate continental monsoon climate. It is dry and windy in spring and hot in summer, with a cold and long winter and scarce precipitation. The annual average precipitation is 180 mm, the annual average wind speed is 5 m·s^−1^, and the frost-free period lasts about 130 d. The soil type is primarily sandy soil with a loose structure and a shallow groundwater table. The low precipitation is distributed unevenly throughout the year. The area has a fragile ecological environment. The zonal vegetation type is Stipa desert steppe.

### 2.2. Experimental Design and Treatment

A completely randomized block design was used in the field experiment, with five treatments and six replicates for a total of 30 plots. The plots measured 5 m × 5 m with an aisle distance of 1 m. The five treatments were long-term nutrient addition, shallow plowing, nutrient addition and shallow plowing, short-term nutrient addition, and a control. Long-term nutrient addition (LN): the same scheme (NPKμ) as the international nutrient addition research network (Nutrient Network) was adopted, i.e., N, P, and K were each 10 g·m^−2^·y^−1^. The microelements were provided by Scotts Micromax compound fertilizer, which was applied only in the first year, and the application amount was 100 g·m^−2^. The specific formulas of nutrient addition measures are shown in Table 1. Shallow plowing (P): After removing all aboveground biomass, a small rotary cultivator was used to plow the top 15 cm, and the belowground biomass was removed (only larger roots needed to be removed). The interaction between plowing and nutrient addition (P + N) was studied, i.e., after the removal of the aboveground biomass, fertilizer was applied immediately, followed by shallow plowing and the removal of large roots. Short-term nutrient addition (SN) was applied with exactly the same conditions as for the long-term nutrient addition, except that the treatment was discontinued after year 5. For the control treatment (C), no measures were taken (Figure 2). In this experiment, because the sampling time was in 2021, the third year of the establishment of the shallow plowing and nutrient addition networking experimental platform had not yet met the conditions required for short-term nutrient addition treatment (i.e., application of the nutrients was stopped in the fifth year). In this study, the short-term nutrient addition treatment (SN) was exactly the same as the long-term nutrient addition treatment (LN) so the two treatments were merged as the nutrient addition treatment (N).

### 2.3. Sample Collection

Soil samples were collected in the summer of 2021 (July) to quantify the variation characteristics of the soil microbial entropy in the three typical grasslands under the shallow plowing and nutrient addition treatments. In the sampling area of each plot (1 m × 1 m), five 3.5-cm-diameter drills with a soil depth of 0–10 cm were randomly taken, and the collected soil samples were mixed after removing the stones and visible roots. The soil samples were sieved using a pore size of 2 mm and divided into two parts. One part was stored in a 4 °C refrigerator and used to determine the soil MBC. The other soil sample was dried naturally, ground, and passed through a 0.25-mm sieve for the determination of the soil indicators.

### 2.4. Analysis Methods

The soil pH was measured by a pH meter; conductivity was measured by a conductivity meter, and the SOC was determined by the potassium dichromate–sulfuric acid heating method. The total nitrogen was determined by the Kjeldahl method. The soil MBC was determined by the chloroform fumigation–K_2_SO_4_ extraction method [27]. The soil microbial entropy was calculated as the ratio of MBC to SOC (MBC/SOC).

### 2.5. Data Analysis

All experimental data were analyzed by SPSS 26.0 software (IBM Corp, Armonk, NY, USA) and Excel 2010 (Microsoft Corp, Redmond, WA, USA). Single-factor analysis (one-way ANOVA) was used to analyze the differences of each index among the different treatments. Principal component analysis (PCA) was used to standardize the multivariate data. On the basis of retaining the original information as much as possible, a dimension reduction of the multivariate data was carried out; the key factors of soil quality were screened out from multiple grassland soil indexes, and their comprehensive scores were calculated.

## 3. Results and Analysis

### 3.1. Soil Physical and Chemical Properties

The soil pH was alkaline in the typical steppe. Compared to the control, the soil pH was significantly decreased by the nutrient addition. The soil pH of the desert steppe was also alkaline, and its value was the highest among the three grassland types. The soil pH decreased significantly under the combination of shallow tillage and nutrient addition. The pH value of the meadow steppe soil was neutral to acidic; compared to the control group, the three treatments had no significant effect on the pH value. In the meadow steppe and typical steppe, shallow plowing, nutrient addition, and their interaction had no significant effect on the soil electrical conductivity, while in the desert steppe, the nutrient addition significantly increased the soil electrical conductivity compared to the control group. Shallow plowing and nutrient addition promoted the accumulation of organic carbon in the typical steppe and desert steppe but had no significant effect on the accumulation of organic carbon in the meadow steppe. The nutrient addition measures increased the total nitrogen content in the meadow steppe soil, while the shallow plowing and nutrient addition measures had no significant effect on the total nitrogen accumulation in the typical steppe or desert steppe soil. In addition, shallow plowing and nutrient addition significantly increased the ratio of SOC to total nitrogen in the three grasslands compared to the control (Figure 3).

### 3.2. Variation Characteristics of Microbial Biomass Carbon Content and Microbial Entropy in Grassland Soil

The variation characteristics of the MBC and microbial entropy were basically the same under the different grassland management measures. In the meadow steppe, compared to the control, the shallow plowing and nutrient addition measures and their interaction had significant negative impacts on the soil MBC and microbial entropy. In the typical steppe, the soil MBC under plowing was significantly higher than that in the control group, and the microbial entropy had no significant difference. For nutrient addition and its interaction with plowing, the soil MBC and microbial entropy were significantly lower than those of the control group. Plowing and the interaction of shallow plowing and nutrient addition significantly increased the MBC and microbial entropy in the desert steppe, while the nutrient addition showed significant negative effects (Table 2 and Table 3).

### 3.3. Changes in Grassland Soil Quality under Different Grassland Management Measures

A PCA of the soil quality under different grassland management measures was carried out. Seven soil indexes were represented by two principal component axes: pH (X_1_), EC (X_2_), SOC (X_3_), TN (X_4_), C/N (X_5_), MBC (X_6_), and qMB (X_7_). In the different grasslands, the two principal components explained 92.478%, 88.900%, and 91.643% of the total variances of the meadow steppe, typical steppe, and desert steppe, respectively, and these were used to evaluate the grassland soil quality (Figure 4, Figure 5 and Figure 6). The matrix formulas for the soil quality score coefficients of different grasslands are presented in Table 4, where y_1_ and y_2_ are the first and second principal components, respectively.

For the comprehensive scores (Table 5), in the meadow steppe, the comprehensive score of the grassland soil quality in the control group was positive, and under the treatments of shallow plowing, nutrient addition, and their interaction, the scores were far lower than those of the control group, and both were negative. In the typical steppe, the comprehensive score of the grassland soil quality in the control group was positive and that with the shallow plowing treatment was higher than that in the control group, while the comprehensive scores of the grassland soil quality with the nutrient addition and its interaction with the shallow plowing treatment were far lower than those in the control group, and both were negative. In the desert steppe, the comprehensive score of soil quality in the control group was negative. The scores of shallow plowing, nutrient addition, and their interaction were higher than those in the control group. The comprehensive score of soil quality in the nutrient addition treatment was the highest and much higher than that in the control group.

## 4. Discussion

### 4.1. Basic Properties of Soil

In the meadow steppe, shallow plowing and nutrient addition had no significant effect on the soil pH. In the typical steppe, nutrient addition significantly reduced the soil pH. There was no significant difference in the soil electrical conductivity between the meadow steppe and the typical steppe under shallow plowing and nutrient addition, a result that is consistent with the results of Yan Ruirui [28]. Shallow plowing and nutrient addition measures promoted the accumulation of organic carbon in the typical grasslands and desert grasslands. The reason was that the nutrient addition measures reduced the loss of SOC to a certain extent. The shallow plowing measures were conducive to the formation of an excellent soil structure; this ensured the normal functioning of the soil carbon cycle and inhibited the release of carbon from the soil [25], consistent with the function of carbon sequestration. Studies have shown that fertilization can improve the SOC content [29], especially the available carbon content [30]. Shallow plowing had no significant effect on the soil total nitrogen accumulation in the typical steppe or desert steppe. Zhang Jie’s study showed that different tillage measures can cause the soil total nitrogen content to decrease, in contrast to the results of this experiment [31].

### 4.2. Changes in Microbial Biomass Carbon and Microbial Entropy in Grassland Soil

Soil microbial entropy can effectively track the state of organic matter after it is added to the soil, and it is a useful indicator for evaluating the dynamics and quality of microbial carbon in the soil. The higher the value, the stronger the activity of the soil organic matter, thus indicating an improvement in the utilization efficiency of microorganisms for organic matter. In this study, the changes in the soil microbial entropy in three typical grasslands under shallow plowing, nutrient addition, and their interaction were compared at the regional scale. Moreover, we found that within the same grassland, the soil MBC and microbial entropy showed parallel change characteristics under different treatments. The results are similar to those of Yang Ning [32]. Due to the influence of the grassland types, the response of the microbial entropy to the shallow plowing and nutrient addition measures in the different grasslands showed clear spatial variation characteristics.

In the meadow steppe, shallow plowing and nutrient addition significantly reduced the accumulation of MBC in the grassland soil, and the microbial entropy also decreased. This is because shallow plowing destroyed the soil surface structure, resulting in some soil microorganisms being exposed to the air and thus losing the protective effect of the soil. The addition of nutrient elements reduced the numbers and activities of soil microbial populations, thereby affecting the MBC content, consistent with the results of Yang Jingyi et al. [33]. In contrast, the meadow steppe has a humid climate, good surface soil permeability, and rich organic matter [34]. The disturbance by human management measures can cause the decline of grassland soil quality [35]. In the typical steppe, the soil MBC of grassland under shallow plowing was significantly higher than that of the control group, primarily because shallow plowing enhanced the soil water storage capacity and increased the soil temperature based on breaking the soil compaction, effects that were beneficial to the growth and metabolism of the microorganisms in the soil and the decomposition of organic matter [36]. In addition, under the nutrient addition measures, the soil MBC and microbial entropy of the desert steppe were significantly lower than those of the control group, consistent with the research results of Song et al. [37] and Zhou et al. [38]. This may have been due to the drought and low rainfall climate of the desert steppe. The soil surface layer was affected by wind erosion and evaporation; its structure was loose, and its water and fertilizer conservation abilities were poor. Thus, even under the condition of artificial nutrient addition, the soil still experienced nutrient loss. Therefore, the dual limitation of water and nutrients made the soil microbial biomass of the desert steppe decrease significantly. Zhang suggested that nitrogen addition caused a decline in the soil microbial biomass [39]. In contrast, under the influence of the interaction of shallow plowing and nutrient addition, the soil MBC and microbial entropy in the desert steppe were significantly higher than those in the control group, possibly due to the fact that under the nutrient addition measures, shallow plowing altered the original loose soil structure of desert steppe soil to a certain extent and enhanced its fertilizer retention ability, and thus, was beneficial for the growth of soil microorganisms [40].

### 4.3. Soil Quality of Grassland under Different Grassland Management Measures

We conducted a PCA on the soil quality under different grassland management measures in the three grasslands. The results show that the shallow plowing and nutrient addition measures significantly decreased the soil MBC and microbial entropy in the meadow steppe, resulting in a lower comprehensive score of soil quality than that in the control group. In the typical steppe, due to the influence of the shallow plowing measures, the soil microbial entropy of the grassland in this area was higher than that of the control group and its two treatments, such that the comprehensive score of soil quality ranked first among the four treatments. In the desert steppe, shallow plowing and shallow plowing + nutrient addition significantly increased the soil microbial entropy in the region. Under the nutrient addition measures, the soil microbial entropy of the desert steppe also showed a downward trend, a result that is consistent with its variation characteristics in the meadow steppe but different from those of the typical steppe. This also indicates that microbial entropy varied in different grassland ecosystems [41]. Some studies have shown that nitrogen addition is conducive to the growth of microorganisms but leads to decreases in the numbers and activities of microorganisms [37]. In addition, it is worth noting that the C/N ratio of grassland soil under the nutrient addition measures was significantly increased. When the microbial entropy was significantly decreased, the comprehensive score of soil quality ranked first among the four treatments. This may have been due to the arid desert steppe having experienced grassland soil degradation [42], resulting in an extremely fragile ecological environment with smaller microbial populations and a much higher abundance of bacterial communities than fungal communities [22]. Therefore, the soil C/N ratio may be a key factor in assessing grassland soil quality in the region. Studies have also shown that low microbial entropy is due to high nutrients at higher C/N ratios, which limit microbial growth [43].

## 5. Conclusions

In this study, the response mechanisms of the soil microbial entropy of three typical grasslands under shallow plowing, nutrient addition, and their interaction were compared at the regional scale, and the soil quality of grasslands under different management measures was comprehensively evaluated. The results show that in the meadow steppe, shallow plowing and nutrient addition significantly decreased the soil microbial entropy of the grassland, and its soil quality was also inferior to that of the control group. Shallow plowing significantly improved the soil MBC and soil quality of the typical steppe; nutrient addition significantly reduced the soil microbial entropy in the desert steppe, but the comprehensive quality of the grassland soil was significantly improved and significantly different from that of the other two grasslands. Based on the results of this experiment, we suggest that in the meadow steppe, the original habitat conditions are excellent, and excessive human interference should be avoided to preserve the condition of the ecological environment. In the typical grassland, appropriate shallow plowing is conducive to improving the overall quality of the soil; this can help to solve the problem of grassland degradation in the region. In the desert steppe, the appropriate application of exogenous nutrients can enhance soil fertility and improve the soil comprehensive quality. Furthermore, shallow plowing measures will also have a positive effect on the improvement of soil quality. Therefore, when developing grassland management measures, we should adapt to the local conditions and make reasonable choices to effectively address the degradation of grassland ecosystems under global climate change.

## Figures and Tables

**Figure 1 ijerph-20-02308-f001:**
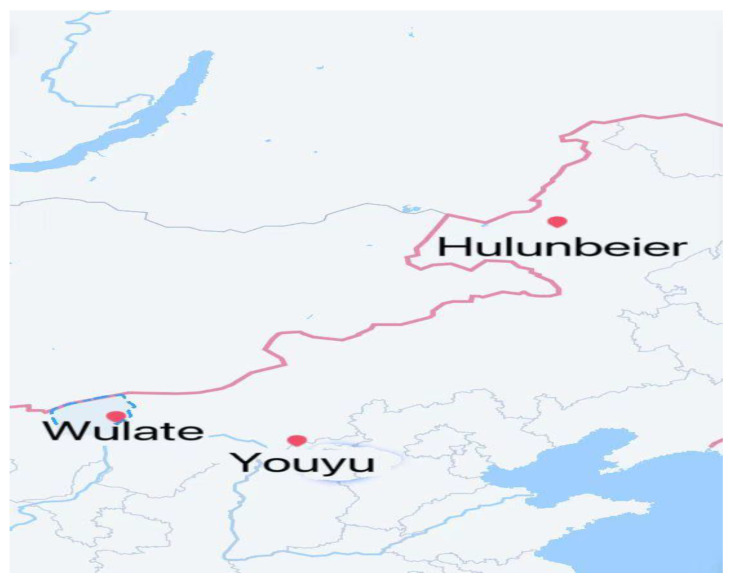
Distribution map of test sample points.

**Figure 2 ijerph-20-02308-f002:**
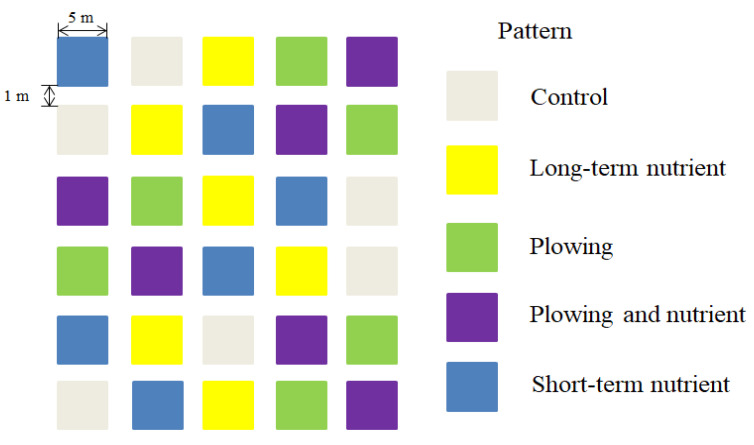
Schematic diagram of experimental design.

**Figure 3 ijerph-20-02308-f003:**
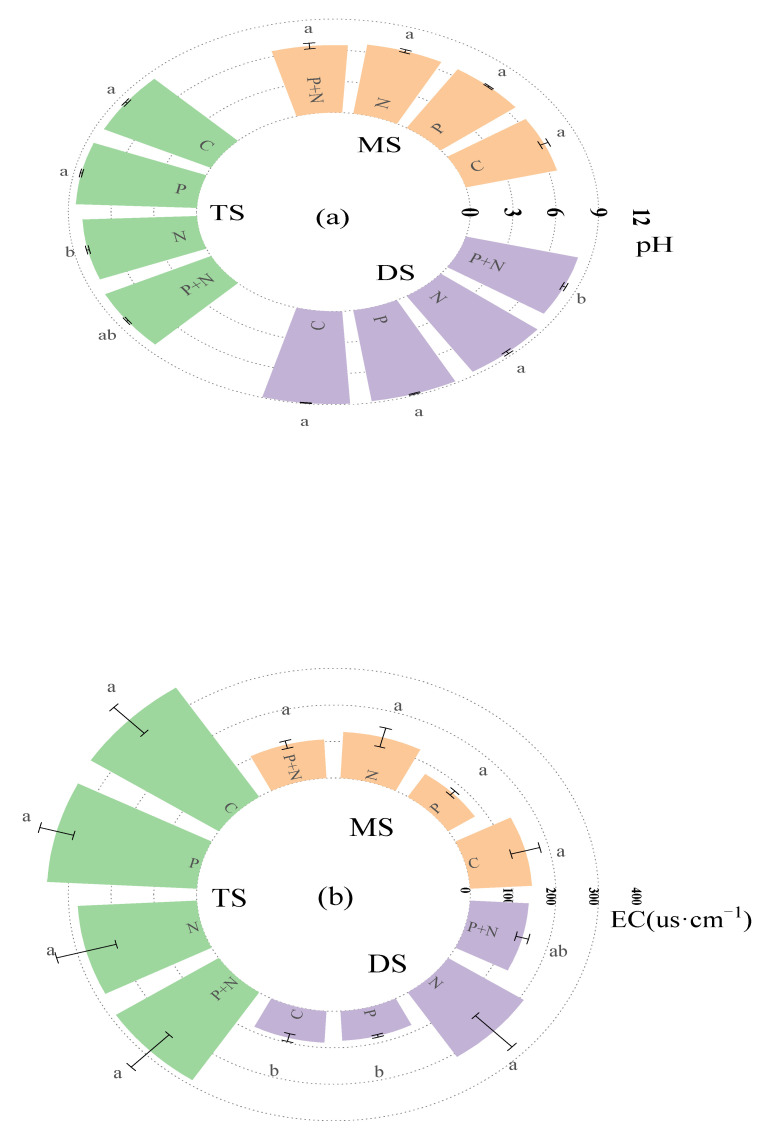
Effects of plowing and nutrient addition on soil basic properties. (**a**) Soil pH; (**b**) soil electrical conductivity (EC); (**c**) soil organic carbon (SOC); (**d**) soil total nitrogen (TN); (**e**) soil C/N ratio. MS: meadow steppe; TS: typical steppe; DS: desert steppe. Different lowercase letters indicate that the same indicator differs significantly among treatments (*p* < 0.05).

**Figure 4 ijerph-20-02308-f004:**
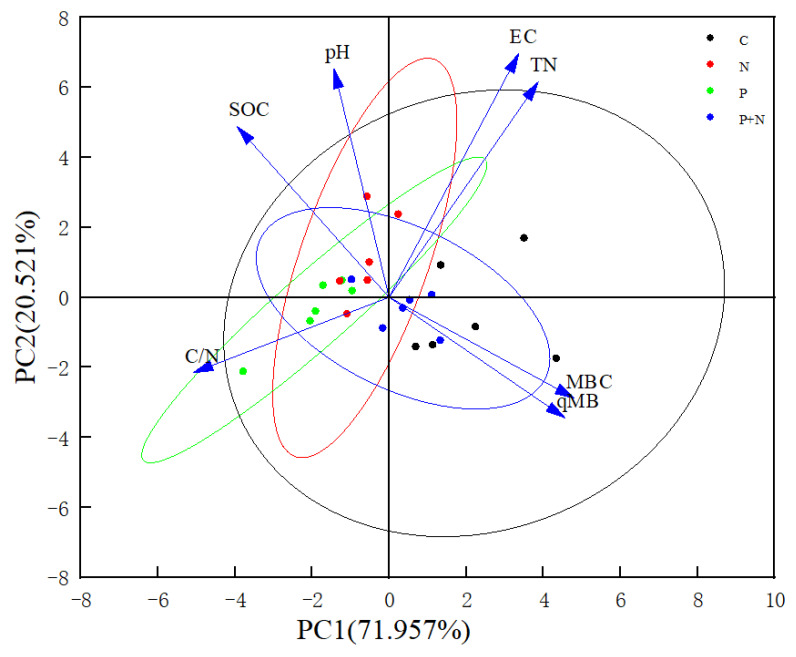
Principal component analysis of meadow steppe. pH: Soil pH; EC: soil electrical conductivity; SOC: soil organic carbon; TN: soil total nitrogen; C/N: soil C/N ratio; MBC: soil microbial biomass carbon; qMB: soil microbial entropy. The same as below.

**Figure 5 ijerph-20-02308-f005:**
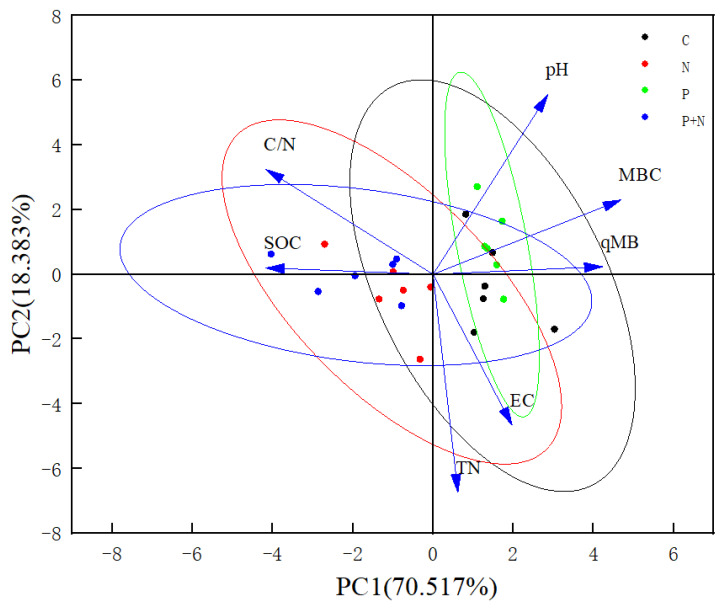
Principal component analysis of typical steppe.

**Figure 6 ijerph-20-02308-f006:**
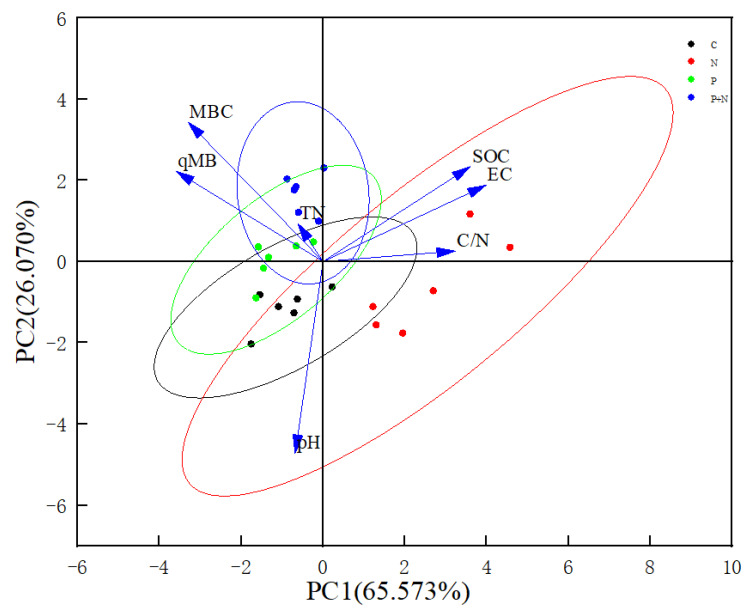
Principal component analysis of desert steppe.

**Table 1 ijerph-20-02308-t001:** Specific formula of the nutrient addition.

Nutrient		Form/Element	Analysis
Major element applied every year in the treatment 10 g·m^−2^·y^−1^	Nitrogen: time release urea	(NH_2_)_2_CO or N_2_H_4_CO	43.00%
	Phosphorus: triple super phosphate	P_2_O_5_	45%
		P (by atomic mass)	19.63%
		Ca	16%
		S	1%
		Mg	1%
	Potassium sulfate (K_2_SO_4_)	K (by atomic mass)	44.9%
		S (by atomic mass)	18%
Microelement applied only in the first year of treatment 100 g·m^−2^	Ca		6%
	Mg		3%
	S		12%
	B		0.10%
	Cu		1%
	Fe		17%
	Mn		2.50%
	Mo		0.05%
	Zn		1%

**Table 2 ijerph-20-02308-t002:** Linear regression analysis of soil microbial biomass carbon.

Coefficient						
Model		Unstandardized Coefficient		Standardized Coefficient	t	Significance
		B	Stderr	Beta		
1 Meadow steppe	(Constant)	0.632	0.003		183.761	0.000
	Plowing	−0.527	0.005	−1.083	−108.327	0.000
	Nutrient	−0.500	0.005	−1.028	−102.788	0.000
	P + N	−0.302	0.005	−0.621	−62.157	0.000
	Control	0				
1 Typical steppe	(Constant)	0.515	0.006		79.475	0.000
	Plowing	0.042	0.009	0.185	4.586	0.000
	Nutrient	−0.147	0.009	−0.648	−16.018	0.000
	P + N	−0.190	0.009	−0.838	−20.722	0.000
	Control	0				
1 Desert steppe	(Constant)	0.305	0.005		63.689	0.000
	Plowing	0.197	0.007	0.558	29.064	0.000
	Nutrient	−0.151	0.007	−0.429	−22.315	0.000
	P + N	0.221	0.007	0.628	32.719	0.000
	Control	0				

Note: Dependent variable: MBC.

**Table 3 ijerph-20-02308-t003:** Linear regression analysis of soil microbial entropy.

Coefficient						
Model		Unstandardized Coefficient		Standardized Coefficient	t	Significance
		B	Stderr	Beta		
1 Meadow steppe	(Constant)	0.018	0.000		37.771	0.000
	Plowing	−0.016	0.001	−1.066	−22.660	0.000
	Nutrient	−0.015	0.001	−1.018	−21.628	0.000
	P + N	−0.009	0.001	−0.631	−13.410	0.000
	Control	0				
1 Typical steppe	(Constant)	0.079	0.004		18.479	0.000
	Plowing	0.003	0.006	0.058	0.422	0.678
	Nutrient	−0.024	0.006	−0.557	−4.022	0.001
	P + N	−0.036	0.006	−0.827	−5.967	0.000
	Control	0				
1 Desert steppe	(Constant)	0.118	0.011		10.516	0.000
	Plowing	0.040	0.016	0.327	2.523	0.020
	Nutrient	−0.077	0.016	−0.629	−4.849	0.000
	P + N	0.034	0.016	0.280	2.157	0.043
	Control	0				

Note: Dependent variable: qMB.

**Table 4 ijerph-20-02308-t004:** Matrix formulas of soil quality score coefficients for different grasslands.

Type	Score Coefficient Matrix Formula
Meadow steppe	y_1_ = −0.269X_1_ + 0.379X_2_ − 0.407X_3_ + 0.295X_4_ − 0.418X_5_ + 0.423X_6_ + 0.422X_7_
	y_2_ = 0.497X_1_ + 0.425X_2_ + 0.291X_3_ + 0.601X_4_ − 0.207X_5_ − 0.205X_6_ − 0.204X_7_
	y = 0.7196y_1_ + 0.2052y_2_
Typical steppe	y_1_ = 0.317X_1_ + 0.433X_2_ − 0.408X_3_ − 0.078X_4_ − 0.375X_5_ + 0.045X_6_ + 0.449X_7_
	y_2_ = 0.407X_1_ + 0.025X_2_ + 0.221X_3_ − 0.743X_4_ + 0.462X_5_ + 0.137X_6_ + 0.011X_7_
	y = 0.7052y_1_ + 0.1838y_2_
Desert steppe	y_1_ = −0.116X_1_ + 0.451X_2_ + 0.449X_3_ − 0.385X_4_ + 0.455X_5_ − 0.289X_6_ − 0.379X_7_
	y_2_ = −0.614X_1_ + 0.013X_2_ + 0.194X_3_ − 0.272X_4_ + 0.118X_5_ + 0.568X_6_ + 0.416X_7_
	y = 0.6557y_1_ + 0.2607y_2_

**Table 5 ijerph-20-02308-t005:** Comprehensive scores of soil quality of different grassland types.

Type	Treatment	y_1_ Score	y_2_ Score	y Score	Rank
Meadow steppe	C	2.73	−0.22	1.92	1
	P	−2.60	−0.88	−2.05	4
	N	−0.74	1.75	−0.17	2
	P + N	−0.60	−0.65	−0.30	3
Typical steppe	C	1.85	−1.24	1.08	2
	P	1.93	1.35	1.61	1
	N	−1.37	−0.55	−1.07	3
	P + N	−2.41	0.44	−1.62	4
Desert steppe	C	−1.86	−1.47	−1.60	4
	P	−0.98	0.39	−0.54	3
	N	3.05	−0.64	1.83	1
	P + N	−0.21	1.71	0.31	2

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
