# Peer review of "Response of Grassland Soil Quality to Shallow Plowing and Nutrient Addition"

_ijerph, 2023, doi:10.3390/ijerph20032308_

Round 1

Reviewer 1 Report

The subject of the role of microorganisms in grassland soils and their response to human activity (cultivation and fertilization) taken up by the authors is up-to-date, especially in relation to climate change.

The manuscript needs a few corrections:

1.       The introduction, especially with regard to soil microbes, is too general.  The authors state that the number and activity of soil microorganisms will change when they are affected by tillage, fertilization and other management measures, but they do not specify how, e.g. depending on the type of soil and other factors. Such studies have been conducted and particular groups of microbes (aerobic, anaerobic) and their interaction with plants were discussed (fungi, mycorrhiza, nodule bacteria, symbiosis, etc.). I think that these issues should be presented in more detail, paying attention to individual groups of microorganisms and their responses to changes in soil factors caused by human activity.

2.       In the Material and Methods chapter, subsection 2.1. types of soil should be described in more detail, because their biological and physicochemical properties provide plants with appropriate conditions for development.

3.       Figure 2 - No legend regarding the colors of individual squares. In addition, the inscriptions in the squares: disturbance, cessation, control - have not been explained.

4.       Subsection 2.2 needs to be supplemented and rewritten in some passages. There is no information on what control (C) is - how fertilized, how used (mowing, grazing). Was fertilization varied depending on the type of soil? Line 129-130 is for N, P and K doses, but only two values are given (10 g/m2), which nutrients are they for?

5.       Subsection 2.4 pay attention to the tense and change the present tense to the past tense (2 times in line 151) and add in brackets the abbreviations MBC and SOC, which are used later.

6.       In the Results chapter, the numbers of subsections (3.2, 3.3) are missing, but their titles are present.

7.       In Figures 4, 5, 6, increase the font size in the descriptions of individual points, it is disproportionately small compared to the size of the figures.

8.       The way of referring to the literature and the list of references is different from that recommended in MDPI journals.

9.       The Discussion section is well written.

10.   Conclusions are in the form of a summary of the results. It would be better to give them in relation to the type of steppe management that best protects the soil.

Reviewer 2 Report

The manuscript «Response of microbial quotient in grassland soil to plowing and nutrient addition» describes a study conducted on experimental sites of the effects of fertilization and shallow plowing of steppe soils.  The results obtained by the authors can contribute to the improvement of soil quality management.

The positive side of the study is the desire to use standard techniques, such as protocols Nutrient Network. Standardization allows you to compare data from different geographical areas, as well as use data for meta-analysis. In this connection, I hope that the authors of the article will be able to provide free access to their data for other researchers.

I want to point out the following shortcomings of the article.

1) It seems to me that the title does not fully correspond to the contents of the manuscript. The article is rather devoted to the study of soil quality. The microbial component only indirectly affects the indicators measured by the authors.

2) The description of the experiment scheme and treatments is confusing and unclear.

Line 125: Five treatments are described. It is not entirely clear why the two types of processing were combined. No statistical differences were found between them?

What are the specific dates for the beginning and end of the experiment? In what season and dates were the treatments carried out?

What equipment was used? What fertilizers were used? Chemicals? Brand?

3) To explain the results obtained, you write several times about changes in the structure of the soil and its moisture capacity. Quantifying these parameters would indeed be important to explain the results. Have such measurements been carried out and are the results available?

Round 2

Reviewer 2 Report

Dear authors of the article. Thank you for the detailed explanations you provided. Those aspects of the article that were incomprehensible to me have become clear.